# Has The Time Arrived to Refine The Indications of Immunosuppressive Therapy and Prognosis in IgA Nephropathy?

**DOI:** 10.3390/jcm8101584

**Published:** 2019-10-02

**Authors:** Bogdan Obrișcă, Ioanel Sinescu, Gener Ismail, Gabriel Mircescu

**Affiliations:** 1Department of Nephrology, Fundeni Clinical Institute, 022328 Bucharest, Romania; gener732000@yahoo.com; 2Department of Uronephrology, “Carol Davila” University of Medicine and Pharmacy, 020021 Bucharest, Romania; umfisinescu@gmail.com (I.S.); gmircescu@hotmail.com (G.M.); 3Center of Uronephrology and Renal Transplantation, Fundeni Clinical Institute, 022328 Bucharest, Romania; 4Department of Nephrology, “Dr. Carol Davila” Teaching Hospital of Nephrology, 010731 Bucharest, Romania

**Keywords:** IgA nephropathy, kidney biopsy, renal survival, immunosuppression, MESTC score

## Abstract

Immunoglobulin A nephropathy (IgAN) is the most frequent glomerular disease worldwide and a leading cause of end-stage renal disease. Particularly challenging to the clinician is the early identification of patients at high risk of progression, an estimation of the decline in renal function, and the selection of only those that would benefit from additional immunosuppressive therapies. Nevertheless, the pathway to a better prognostication and to the development of targeted therapies in IgAN has been paved by recent understanding of the genetic and molecular basis of this disease. Merging the data from the Oxford Classification validation studies and prospective treatment studies has suggested that a disease-stratifying algorithm would be appropriate for disease management, although it awaits validation in a prospective setting. The emergence of potential noninvasive biomarkers may assist traditional markers (proteinuria, hematuria) in monitoring disease activity and treatment response. The recent landmark trials of IgAN treatment (STOP-IgAN and TESTING trials) have suggested that the risks associated with immunosuppressive therapy outweigh the benefits, which may shift the treatment paradigm of this disease. While awaiting the approval of the first therapies for IgAN, more targeted and less toxic immunotherapies are warranted. Accordingly, the targeting of complement activation, the modulation of mucosal immunity, the antagonism of B-cell activating factors, and proteasomal inhibition are currently being evaluated in pilot studies for IgAN treatment.

## 1. Introduction

Immunoglobulin A nephropathy (IgAN), first described in 1968 by Jacques Berger [1], is the most frequent primary glomerular disease worldwide diagnosed by kidney biopsy [2]. The most recent survey evaluated over 40,000 kidney biopsies across four continents and reported the frequency of IgAN to be 22% and 39% of all glomerular diseases in Europe and Asia, respectively [3]. 

The clinical presentation of IgAN is extremely variable, ranging from asymptomatic microscopic hematuria to a rapidly progressive course or nephrotic syndrome. The 10-year renal survival rate reported by different registries varies between 60% and 95%, while up to 50% may reach end-stage renal disease (ESRD) within 20 years of diagnosis [4,5,6]. Moreover, even those patients initially considered to have a “benign” disease show a progressive decline of renal function if followed for more than 20 years [7]. Regional environmental factors, differences in racial composition, and genetic susceptibility are important contributors to IgAN epidemiology and might explain disease susceptibility, heterogeneous presentation, and the risk of progression [8]. Additionally, the spectrum of renal lesions parallels the clinical findings, ranging from normal glomerular appearance to severe proliferative changes [9]. It is important to note that in the majority of cases, the disease starts at an early age with few manifestations and follows a relentlessly progressive clinical course [10], which is an important challenge in countries that lack screening programs where IgAN may remain undiagnosed or is captured at an advanced stage of disease when a particular treatment will not be of benefit [1,10].

Particularly challenging to the clinician is not only the early identification of patients at high risk of progression, but also the accurate estimation of the decline in renal function in order to select only those that would benefit from a specific therapy, before the irreversible loss of renal tissue occurs [11].

Current clinical guidelines [12] assess the risk of disease progression based on classical risk factors (proteinuria, renal function at renal biopsy and during follow-up) and recommend corticosteroids only in patients with persistent proteinuria and relatively preserved renal function, albeit with a low quality of evidence (2C). These recommendations are based on previous randomized clinical trials, which were criticized for the inconsistent use of a renin-angiotensin system (RAS) blockade and the inadequate reporting of adverse events [13,14,15,16]. Nevertheless, there is additional solid evidence from retrospective, observational studies of the benefits of immunosuppressive therapy in IgAN, especially in those patients with severe proteinuria and with altered renal function [17]. However, the current role of immunosuppression in IgAN is strongly debated [13]. Recent major randomized clinical trials raise concern about the serious side effects of immunosuppressive treatment, mostly regarding infections, with “apparently” no additional benefit to that of conservative therapy in terms of renal survival [18,19]. Despite this strong data, concerns have been raised whether the short follow-up period of those recent trials is sufficient to draw any definitive conclusions with respect to renal survival in such a slowly progressive disease and whether the benefits of immunosuppressive therapy will become evident after a longer observation period, similarly to past clinical trials [20,21,22]. Therefore, many questions about the utility, optimal regimen, and duration of immunosuppressive therapy still remain to be answered [18,19].

The scope of this review is to provide an update of the current view of pathogenesis, prognosis, and treatment options of IgAN. The high prevalence and the progressive nature of this disease make it an important contributor to the global burden of chronic kidney diseases, therefore justifying the imperious need to search markers of disease progression and newer therapies to mitigate the decline in renal function with fewer side effects than the current, nonspecific immunosuppression [1,11]. Finally, an improved treatment algorithm is proposed with the intention to individualize the management of such patients.

## 2. Transitioning from Pathogenesis of IgA Nephropathy to Prognosis and Treatment of IgA Nephropathy

The pathway to a better prognosis and to the development of targeted therapies in IgA nephropathy has been paved by recent advances in understanding its pathogenic mechanisms. The hallmark of IgAN is the mesangial deposition of polymeric IgA1 and IgA1-immune complexes. The autoimmune process in IgAN (extensively reviewed previously in References [23,24,25,26]) can be synthesized in a “multihit” mechanism: a genetically determined increased production of circulating galactose-deficient IgA1 (Gd-IgA1), an antiglycan antibody response, the formation of circulating immune complexes with subsequent glomerular deposition, the activation of mesangial cells, and glomerular injury (Figure 1).

Despite the restricted deposition of immune complexes in the mesangial area, all renal compartments are injured in IgAN. Murine and human studies have demonstrated that higher Gd-IgA1 serum levels are associated with a stronger mesangial cell response, which translates into more severe glomerular and tubulointerstitial lesions [27,28,29]. Mesangial cell-derived inflammatory and fibrotic mediators (IL-6, MCP-1, TGFβ, TNFα) lead to podocyte dysfunction and their progressive loss and to tubulointerstitial injury [27,28,29]. This mesangial–podocyte–tubular crosstalk provides the rationale for the spread of initial mesangial injury to the entire nephron and ultimately to progressive loss of renal function [29].

The polymeric nature of IgA1 of mucosal origin in the mesangial deposits, the coexistence of pharyngitis or other mucosal infections with gross hematuria, and the potential role of dietary antigens or microbial products in disease pathogenesis have long suggested a gut–renal connection in IgAN [10,26,30]. The interaction of dietary or microbial antigens with the mucosal immune system results, through T-cell-dependent or -independent mechanisms, in the overproduction of B-cell activating factors (BAFF – B cell activating factor, APRIL – a proliferation inducing ligand) that promote B-cell proliferation and class switch from IgM to IgA1 [30]. Thus, a better understanding of the role of mucosal immunity, B-cell activity, and alternative and lectin pathways of complement activation could offer insights into potential therapeutic targets to be tested in future trials [1].

A better understanding of the IgAN disease process permits the targeting of an earlier pathogenic event (autoimmune process–renal lesion–clinical manifestation). Predicting IgAN outcome currently relies on clinical data collected at the moment of kidney biopsy and during follow-up [11]. However, clinical data (proteinuria, hypertension, renal function) are only the late manifestations of a pathogenic process that starts many years before the disease becomes clinically manifest and expresses an already established renal lesion. Nevertheless, the role of histological data added to baseline clinical features for an earlier assessment of risk is emerging, as is the role of potential biomarkers to monitor disease activity and treatment response, which could overcome these difficulties in patient management [11].

However, as all of the accepted prognostic factors have been validated in retrospective cohorts, they could be biased by the heterogeneous treatment interventions. The main difficulty in designing a controlled, prospective study is the smoldering course of IgAN, which requires many years of prospective observation before a risk factor may become apparent.

Despite current treatment strategies that merely slow disease progression, by transitioning from pathogenesis to bedside, the future perspective of IgAN is targeted toward the individualized management of patients.

### 2.1. Traditional Risk Factors for Disease Progression

Multiple clinical and demographic variables have emerged as potential predictors of renal survival in IgAN over the past decades [31]. The most robust indicators of renal function decline are estimated glomerular filtration rate (eGFR), hypertension, and proteinuria (both at baseline and during follow-up) [31,32]. Time-averaged proteinuria (TaP) has a different significance in IgAN in contrast to other glomerular diseases [33,34]. A gradual decline in renal function parallels the increase in TaP, as opposed to membranous nephropathy [35] or focal segmental glomerulosclerosis (FSGS), where proteinuria becomes significantly associated with worse renal survival only in the nephrotic range. Furthermore, regardless of peak levels of proteinuria, a decrease to <1 g/day has been associated with improved renal survival [33]. Additionally, proteinuria reduction has been identified as the most reliable surrogate end point to assess a treatment’s effect on progression to ESRD and is used as an instrument for the accelerated approval of therapies in IgAN [36]. Albeit inconsistently associated with a worse outcome, a recent retrospective study identified patients with persistent microscopic hematuria having a 2.8-fold higher risk of progression [37]. More controversial risk factors are hyperuricemia, gender, and age [31].

### 2.2. Oxford Classification of IgA Nephropathy: Where Do We Stand Today?

Since IgAN does not exhibit a specific serologic profile, a percutaneous kidney biopsy remains the definitive tool to establish the diagnosis of IgAN [11,38]. Additionally, in the past decade, the prognostic value of histological data has become increasingly recognized.

Since its description in 1968, several histologic classifications have been proposed (such as the Lee classification and the Haas system), all based on expert opinions, with a lack of adequate histopathological definitions and high interobserver variability [39]. These classifications have raised concerns similar to those of the International Society of Nephrology/Renal Pathology Society (ISN/RPS) Classification of lupus nephritis: multiple lesions with different prognostic significance were used to define a single class/grade, wrongly assuming that the prognosis within classes was equal irrespective of the histological lesions encountered [40]. Accordingly, their prognostic utility for renal outcomes was inferior to classification based on clinical data and, consequently, did not gain general clinical acceptance [39].

As IgAN is characterized by a diversity of glomerular and tubulointerstitial lesions [38], a group of pathologists and nephrologists undertook an evidence-based approach to identify those lesions that are clinically meaningful and reproducible and could accurately predict renal outcomes [41,42].

The initial validation cohort consisted of 265 patients, both adults and children, across eight countries worldwide and identified four histological lesions as both highly reproducible among pathologists and independently associated with renal outcome: mesangial hypercellularity (M), endocapillary hypercellularity (E), segmental sclerosis (S), and tubular atrophy/interstitial fibrosis (T) [42]. As such, in 2009, the Oxford Classification of IgAN (MEST score) was proposed. It must be emphasized that the initial validation cohort excluded patients at both ends of the clinical spectrum of IgAN: those with an apparently “benign” clinical course (proteinuria <0.5g/d) and those with advanced renal failure and a rapidly progressive course (crescentic disease) [42].

In the past decade, the Oxford Classification has been validated in over 30 retrospective studies comprising more than 9000 patients, across the entire spectrum of clinical manifestation (Figure 2, Table A1). The T score was the most consistent lesion validated and the strongest indicator of poor renal outcome, independent of clinical data, while the other lesions were predictive in some, but not all cohorts [43]. These inconsistencies seemed to be related to the retrospective nature of the validating studies, which, rather than reflecting the natural history of the disease, mainly reflected the local policies of IgAN management, i.e., the variable usage of steroids and other immunosuppressive agents (Table A1) [39].

This is most evident in the case of proliferative lesions, endocapillary and extracapillary hypercellularity. In most validation studies, patients presenting with these type of lesions were more likely to receive immunosuppressive (IS) therapy [39,42]. Therefore, it is not surprising that the E score was found to be predictive of outcome only in those not receiving IS therapy, indirectly suggesting that proliferative lesions are treatment-responsive [39]. Nevertheless, there were two studies that confirmed the prognostic value of the E score in immunosuppression-naïve patients [44,45], although it should be emphasized that the E score remained predictive of outcome irrespective of IS therapy in two other cohorts of patients with Henoch–Schonlein vasculitis [46,47].

Although in the initial studies the presence of crescents was not found to be predictive of renal outcome, mainly due to the low prevalence and exclusion of patients with severe renal impairment, their role was reassessed by Haas et al. [73] in a pooled cohort of 3096 patients from the four largest validation studies (Oxford, VALIGA, and two large Asian cohorts from Japan and China). They identified crescents in 36% of patients, with 61% having less than 10% and 91% less than 25% of glomeruli affected by crescent formation [73]. Despite the low prevalence, after multivariate analysis, a higher risk of the combined event was seen in patients not receiving IS therapy with any proportion of crescent formation (hazard ratio (HR) of 1.51, 95% confidence interval (CI): 1.13–2.02), while the presence of more than 25% of glomeruli with crescents predicted a worse outcome, irrespective of IS therapy (HR 2.29, 95% CI: 1.35–3.91) [73]. Based on these findings, the 2016 Oxford Classification update proposed the addition of a C score to the original MEST score, with C0 (no crescents), C1 (crescents in <25% of glomeruli), and C2 (crescents in ≥25% of glomeruli) [74]. Although validation studies have confirmed the clinical significance of the MEST-C score, there is an urgent need for it to be validated in a prospective, controlled cohort of patients.

The Oxford Classification of IgAN is gaining popularity within the nephrology community because of its ability to predict the long-term outcome. However, how accurately a histological lesion may predict susceptibility to disease progression decades after disease onset remains a matter of debate. It must be emphasized that these lesions are not static and, either spontaneously or with treatment, may change over time. Several repeat biopsy studies [75,76,77,78,79] have consistently shown the reversal of active lesions (endocapillary hypercellularity, fibrinoid necrosis and crescents, mesangial hypercellularity) following IS therapy. As such, active lesions (M, E, C) might resolve or progress to chronic lesions (S, T). This might explain the heterogeneity of lesion frequency seen between validation studies (Table A1) and may be related to the age of the study cohort, baseline characteristics, the threshold for kidney biopsy, and previous treatment interventions. A post hoc analysis of the STOP-IgAN trial [80] evaluated the renal outcome of patients according to baseline histology. Among other findings, the authors showed a disproportionate lower frequency of active lesions (M1: 26%; E1: 17%) compared to chronic lesions (S1: 91%; T1/2: 41%). These data mirrored the time elapsed from initial kidney biopsy and trial enrollment, which ranged from 6.5 to 95 months, with 6% of patients being biopsied more than 3 years before study entry. This analysis underscores the dynamics of renal lesions in IgAN and may explain the limited efficacy of IS seen in this trial. Nonetheless, an updated analysis of the VALIGA cohort [81], with a follow-up period extending up to 35 years, confirmed the value of kidney biopsy findings as independent predictors of the progression risk to ESRD.

Immunohistochemical studies, which were not included in the original Oxford Classification, seem to provide additional prognostic information in terms of IgG codeposition, the location of glomerular immune deposits (restricted to the mesangial area versus spreading to the glomerular capillary wall), and complement deposition. IgG codeposition and the presence of both mesangial and capillary wall deposits are associated with more proliferative lesions (mesangial and endocapillary hypercellularity, segmental necrosis and crescents), but only the spreading of mesangial immune deposits to the peripheral capillary wall was significantly associated with a worse outcome [48,82,83]. Additionally, Espinosa et al. [84] identified that 38.5% of the patients had glomerular deposits of C4d, and their presence was independently associated with a 2.45-fold higher risk of reaching ESRD.

These immunohistochemical findings provided additional insight into IgAN pathogenesis. Capillary wall deposits could recruit additional inflammatory cells and induce more proliferative lesions. Although associated with a worse outcome, this effect was blunted by immunosuppression (IS), which suggested the possibility of an interaction between therapy and immune deposit location [83]. Moreover, C4d deposition in glomeruli could indicate lectin pathway activity, which translates into intensive complement activation and more severe histological lesions.

### 2.3. Novel and Emerging Tests

Although renal damage assessed by kidney biopsy is one of the strongest predictors of renal outcome, there is a constant need to identify reliable noninvasive biomarkers to monitor disease activity and treatment response. The main areas of applicability in the management of patients with IgAN would be earlier risk stratification and a more accurate monitoring of disease activity, allowing for earlier treatment interventions before irreversible renal damage occurs.

Galactose-deficient IgA1 (Gd-IgA1) deposition in the mesangium was shown to be specific for IgAN and IgA vasculitis, while it was not encountered in other glomerular disorders (such as lupus nephritis of membranous nephropathy) [85]. Moreover, given that the sole presence or deposition of Gd-IgA1 is not sufficient to initiate tissue injury, corresponding IgG antiglycan antibodies were shown to colocalize with Gd-IgA1 in glomerular deposits of patients with IgAN, thus confirming their essential role in IgAN pathogenesis [86]. As such, serum levels of Gd-IgA1 and antiglycan antibodies seem to be logical candidates to assess disease activity and have been shown to be increased in IgAN patients, in addition to being related to renal outcome [11]. Maixnerova et al. showed that a higher level of Gd-IgA1 and a greater degree of galactose deficiency were associated with the rate of eGFR decline and poor renal survival [87]. Moreover, Chen et al. identified, in a large cohort of 1210 patients, the Gd-IgA1/C3 ratio as being independently associated with chronic kidney disease (CKD) progression (hazard ratio: 2.03; 95% CI: 1.25–3.29) [88]. Podocyte damage and loss are known to be associated with progressive glomerulosclerosis, and podocyturia was reported to increase in patients with IgAN, even to a greater extent in those with an S1 score [89].

Genetic contribution to the risk of IgAN progression is being increasingly recognized, and genome-wide association studies have identified at least 20 susceptibility loci [90]. In the most recent study, Shi et al. [90] identified a four-single-nucleotide polymorphism risk score that predicted the progression of IgAN and improved the prognostic performance of clinical and clinicopathological risk models.

Since in such a heterogenous disease, prognosis cannot be predicted based only on one variable, incorporating multiple variables into risk scores seems to be the most reliable method. Barbour et al. [91] analyzed a pooled cohort of patients (from the Oxford, North America, and VALIGA studies) and showed that adding the MEST score to clinical data at biopsy (eGFR, proteinuria, and MAP) predicted the renal outcome as well as a two-year follow-up of clinical data did. Additional risk scores of progression have been proposed in different cohorts but are still awaiting validation in independent, ideally prospective cohorts [49,92]. Recently, a large international collaboration proposed a new risk prediction tool that incorporates several clinical and pathological variables (age, ethnicity, eGFR/mean arterial pressure/proteinuria at biopsy, the MEST score, and use of either RAS blockade or immunosuppression at biopsy) [93]. This score was shown to predict more accurately the risk of renal progression (50% decline in eGFR or ESRD) and was developed and externally validated in a pooled cohort of almost 4000 patients.

## 3. Treatment of IgA Nephropathy

IgAN is an important cause of ESRD and is associated with a notable reduction in life expectancy, but it still awaits the approval of disease-modifying therapies [36,94]. The treatment of IgAN has been a major focus of debate over the past decades. Although outdated and currently subject to revision, the Kidney Disease Improving Global Outcomes (KDIGO) guidelines [12] restrict the use of corticosteroids only to those with persistent proteinuria and relatively preserved renal function, and they suggest avoiding other immunosuppressive agents. Since their release in 2012, the results of several randomized clinical trials have seriously questioned the initial guideline recommendations.

### 3.1. Corticosteroids

The story of corticosteroids began in the 1980s with several randomized controlled trials with a small number of patients, a short follow-up period, and initially inconclusive results [1]. However, in the late 1990s, Pozzi et al. [16,95] showed that a 6-month corticosteroid regimen (methylprednisolone pulses in months 1, 3, and 5, followed by oral prednisone) improved renal survival: the serum creatinine doubled after 7 years in 2% of patients in the steroid group compared to 30.2% in the conservative therapy arm. Additionally, steroid treatment was associated with a significant reduction in proteinuria persisting 7 years after initial corticosteroid therapy, whereas it remained unchanged in the control group. This “legacy” effect of corticosteroids on long-term renal survival prompted the introduction of the “Pozzi regimen” into current clinical practice.

In the following years, another four studies compared the efficacy of oral prednisone to dipyridamole, placebo, or omega 3 fatty acids, with mixed, but generally positive results [96,97,98,99]. More recently, two additional trials, one with a long follow-up period (96 months (Manno et al. [15])) and another with a smaller sample and a 48-month follow-up (Lv et al. [14]) further supported the utility of steroids in long-term renal survival, without serious steroid-related adverse effects.

The results of studies investigating the effects of corticosteroids on outcome in IgAN patients with persistent proteinuria and near-normal renal function have been evaluated in some meta-analyses. In one meta-analysis, in which the prognostic utility of “early” proteinuria reduction (9 months) on outcome (doubling of serum creatinine level, ESRD, or death) was evaluated, early proteinuria reduction was associated with a significant improvement in outcome only in the case of corticosteroid therapy, but not in the case of RAS blockade: 29% (95% CI, 6–53%) versus 11% (95% CI, 19–41%) [100]. Another meta-analysis of seven studies concluded that corticosteroids effectively improved kidney survival and reduced proteinuria in patients with IgAN (HR, 0.2; 95% CI, 0.1–0.39) [101]. To note, in the latter meta-analysis, only gastrointestinal side effects were three-fold more frequent in the steroid arm than in the control arm (HR, 2.91; 95% CI, 1.25–6.77), and diabetes mellitus and weight gain were rare, while the infection rate was not even mentioned. Recently, in an analysis of nine studies of IgAN with proteinuria over 1 g/day and normal kidney function, relatively high-dose and short-term steroid therapy (prednisone >30 mg/day or high-dose intravenous methylprednisolone with a duration <1 year) produced significant renal protection, whereas low-dose, long-term steroid use did not. Steroid therapy was associated with a 55% higher risk of adverse events (mostly cushingoid features), but not infection [102].

Accordingly, these initial trials on corticosteroid efficacy in IgAN, although they had generally positive results, were criticized for their lack of a standardized conservative treatment, their inconsistent use of RAS blockade, and the inconsistency of reports on adverse effects (Table 1).

Thus, at the end of 2009, corticosteroid therapy in moderate doses (5–20 mg/day) for a period of 6–96 months seemed to be effective in patients with IgAN with normal renal function (eGFR over 60 mL/min) and moderate proteinuria (1–3 g/day). Cushingoid features, weight gain, and glucose intolerance/new-onset diabetes mellitus or infection were not prominent side effects.

These conclusions changed as the results of two randomized, controlled studies—with a large number of participants, some with eGFRs lower than 60 mL/min and with proteinuria between 1.2 and 2.4 g/day—were published [18,19].

The “Intensive supportive care plus immunosuppression in IgA nephropathy” trial (STOP-IgAN by Rauen et al. [18]) included 337 patients from 32 centers in Germany. After a 6-month run-in period of intensive supportive care (salt restriction, smoking and nonsteroidal anti-inflammatory agent avoidance, blockers of RAS to lower blood pressure to a target below 125/75 mmHg, and statins to control blood lipids), proteinuria decreased below 0.75 g/day in 34.5% of patients, and the remaining 162 patients were subsequently randomized to receive additional IS therapy on top of supportive care or supportive care only. After 36 months, 5% of patients in the supportive care group and 17% in the immunosuppressive group reached the primary end-point of full clinical remission (OR 4.82; 95% CI, 1.43–16.3; *p* = 0.01). However, the proportion of patients who reached the second primary end-point (at least a 15 mL/min decrease of eGFR from baseline) was similar in both arms (OR 0.89; 95% CI, 0.44–1.81; *p* = 0.76). Thus, the clinical benefit was doubtful, as the clinical remission was not accompanied by a better-preserved kidney function. More importantly, the authors stated that “more adverse effects were observed among the patients who received immunosuppressive therapy”, although there was only a trend toward a higher frequency of infections in the IS arm (174 vs. 111; *p* = 0.07), while impaired glucose tolerance/diabetes mellitus and weight gain were significantly more frequent in the IS arm (1/80 vs. 9/82, *p* = 0.02; respectively 5/80 vs. 14/82, *p* = 0.05). However, one patient in the IS arm died because of infection. A subsequent post hoc analysis of the STOP-IgAN trial showed that the difference in full clinical remission in favor of IS therapy was mainly driven by the steroid arm, the decrease in proteinuria being the most important contributor [118]. Again, the high frequency of adverse events was underlined, but at this time without an evaluation of statistical significance [18,118]. Accordingly, corticosteroids plus supportive care was considered to have an inferior risk/benefit ratio to intensive supportive care alone in patients with IgAN, preserved kidney function (eGFR > 60 mL/min), and moderate proteinuria (1–3 g/day).

Subsequently, the Therapeutic Evaluation of Steroids in IgA Nephropathy Global (TESTING) trial [19], designed to be the largest IgAN trial (with a target of 750 recruited patients), aimed to definitively establish the efficacy and safety of steroids, but it was prematurely stopped after 262 patients were randomized due to an excess of serious adverse events (risk difference 11.5%; 95% CI, 4.8–18.2%): mostly infections (risk difference, 8.1%; 95% CI, 3.5–13.9%; *p* < 0.001), including two deaths, in the steroid arm. However, the risk of reaching the primary end-point (ESRD, death due to kidney failure or a 40% decrease in eGFR) was lower after 2 years of follow-up in the corticosteroid arm (risk difference 10%; 95% CI, 2.5–17.9%; *p* = 0.02). Notably, in this study, relatively higher doses of corticosteroids were used for a medium length of time (methylprednisolone 0.6–0.8 mg/kg/day for two months, with subsequent weaning over 4–6 months). Again, the risk reduction (10%) was counterbalanced by the high risk of severe adverse events (11.5%).

Thus, these two trials argued against the efficacy and safety of corticosteroids for IgAN. However, some limitations should be highlighted. First, STOP-IgAN had a 3-year follow-up trial, which could be too short for the long course of IgAN. The annual decline in eGFR in the supportive care group was 1.6 mL/min/year, much lower than in previously reported trials (6.3 mL/min/year [16], 6.2 mL/min/year [15], and 6.9 mL/min/year [14]), which might conceal the differences in short-term outcomes. Second, the STOP-IgAN trial was designed before the proposal of the Oxford Classification of IgAN, and therefore it lacked any histological assessment. We have learned from the Oxford Classification validation studies (Table A1) that histologic features lose their predictive value in patients receiving IS therapy, while repeat biopsy studies have shown the reversal of active lesions upon treatment [75,76,77]. Additionally, these retrospective studies captured the real-world management of these patients, and it was shown that patients with active lesions and even those with severe renal impairment (eGFR down to 30 mL/min) were more likely to be exposed to immunosuppression [17]. In a retrospective propensity score analysis of the VALIGA cohort, Tesar et al. [17] showed that the efficacy of steroid treatment was more evident in those with an eGFR below 50 mL/min, and it increased proportionally with the level of proteinuria. As such, the story of steroid treatment in IgAN should not be abandoned. The validation studies of the Oxford Classification taught us that a histologically driven selection of patients for IS treatment (active vs. chronic lesions) could overcome these issues, and this approach awaits validation in prospective clinical trials (Figure 3).

Insight into the gut–renal connection in IgAN suggests a different approach to steroid treatment [30]. An enteric targeted-release formulation of budesonide was tested in phase 2a [123] and 2b [105] trials, with beneficial effects on proteinuria and renal function decline and fewer side effects, thus making it a future therapeutic agent. Moreover, oral budesonide, the gastro-resistant and pH-modified formulation that has been previously licensed for use in mild-to-moderate active Crohn’s disease, was tested in a small study of patients with IgAN and a high risk of progression, and it improved proteinuria and hematuria and stabilized renal function over a period of 18 months [121]. Recently, an alternative attempt to modulate mucosal-associated lymphoid tissue Toll-like receptor signaling by hydroxychloroquine was tested in a randomized controlled trial [122]. In this trial, hydroxychloroquine reduced proteinuria by almost 50% after 6 months of treatment compared to an increase of 10% in the placebo group [122]. Ultimately, a tonsillectomy was proposed as an alternative procedure for IgAN management with discordant results, with a more favorable outcome being observed in Asia and in association with steroids [124]. As such, current KDIGO guidelines do not recommend a tonsillectomy for IgAN [12].

### 3.2. Immunosuppressants

Other immunosuppressants have yielded conflicting results in randomized clinical trials (Table 1). Ballardie et al. [117] tested the efficacy of a 2-year regimen consisting of cyclophosphamide (1.5 mg/kg/day for 3 months) followed by azathioprine (1.5 mg/kg/day for a minimum of 18 months), in addition to corticosteroids, in a high-risk subgroup of IgAN patients (with a serum creatinine of at least 1.5 mg/dL and a declining renal function of more than 15% in the year preceding study entry). The slope of the eGFR decline and 5-year renal survival were significantly improved by the combination regimen (72% vs. 5% in the placebo arm). This regimen was further evaluated in the STOP-IgAN trial in patients with reduced eGFR (30–59 mL/min/1.73m^2^), where it did not improve the clinical remission rate or renal survival [118]. However, the differences in baseline characteristics of the patients in these two trials could account for the contradictory results. The STOP-IgAN trial excluded patients with a progressive decline in renal function and included patients with lower baseline proteinuria (2.2 g/d vs. 4.2 g/d in the Ballardie trial), while the follow-up period was significantly shorter (36 vs. 72 months). Similarly to corticosteroid trials for IgAN, differences in study design and study populations make direct comparisons of clinical trials and drawing a definitive conclusion difficult.

Mycophenolate mofetil (MMF) has been tested in several trials in IgAN, with heterogeneous treatment regimens (either in dose or duration of treatment), highly variable follow-up periods, and inconsistent results (Table 1). Nevertheless, Tang et al. [108] showed that a 6-month course of MMF (1.5–2 g/d, depending on body weight) was associated with an increased rate of partial remission, and after up to 6 years of follow-up, fewer patients reached the composite endpoint (doubling of serum creatinine/ESRD) compared to the placebo arm (15% vs. 50%). This is another argument supporting the “legacy effect” of IS therapy in IgAN: initial differences in surrogate endpoints can translate into significant differences in hard endpoints (such as renal survival) with prolongation of the follow-up period. Although current KDIGO guidelines suggest that MMF not be used for IgAN treatment based on the low quality of evidence, recent data suggest that at least a subset of patients would benefit from MMF therapy. Beckwith et al. [77] reported a series of 18 patients with endocapillary hypercellularity in a kidney biopsy that were treated with MMF monotherapy. After a median of 24 months, patients underwent a repeat kidney biopsy that showed a statistically significant reduction in the mean percentage of glomeruli with proliferative lesions (endocapillary hypercellularity or crescents). Additionally, Hou et al. [78] showed that in patients with active proliferative lesions, MMF in association with a low-dose steroid regimen had the same efficacy as a full-dose steroid regimen in achieving complete remission and preserving renal function, but with fewer side effects. These latter studies emphasized the missing element from past clinical trials and provided evidence for how to incorporate the MEST-C score into a treatment stratification tool.

Calcineurin inhibitors (CNIs) have been evaluated in several small trials that have suggested a possible renal benefit (Table 1), but all had a short follow-up period and the reduction of proteinuria as a surrogate endpoint. High-quality trials are needed before CNI can be recommended in clinical practice. Azathioprine did not provide additional benefits when associated with corticosteroid treatment [115], while it increased the frequency of adverse events [125].

The B-cell-depleting agent rituximab was an appealing therapeutic agent for IgAN. Lafayette et al. [116] showed that rituximab (1 g given 2 weeks apart) did not alter the level of proteinuria, nor did it improve renal function over 12 months of follow-up. Moreover, despite effectively depleting B-cells, rituximab did not alter the serum levels of IgAN biomarkers (galactose-deficient IgA1 or antiglycan antibodies). Recent data suggest that germinal centers within Peyer patches contain a different subset of B-cells resistant to rituximab (CD20^-^CD19^+^CD27^+^) [26].

### 3.3. Future Directions

A better understanding of the molecular basis of IgAN led to the development of novel therapeutic agents. The targeting of complement activation (eculizumab, the anti-C5a receptor inhibitor avacopan, inhibitors of mannose binding lectin (MBL)-associated serine proteases 1 and 2), the modulation of mucosal B-cell programming by BAFF/APRIL antagonism (blisibimod and atacicept), and proteasomal inhibition (bortezomib) are under evaluation for safety and efficacy in pilot studies of IgAN [126].

In addition, confirming the value of histological data and risk scores in terms of predicting long-term outcome, risk stratification, and treatment individualization awaits validation in a prospective setting, while incorporation into clinical trials may aid in patient recruitment.

## 4. Conclusions

Our understanding of the genetic and molecular basis of IgAN has markedly improved over the last decades. By transitioning from bench to bedside, we face the opportunity to better characterize IgAN patients in terms of clinical manifestations and long-term prognosis, to accurately identify those at high risk of progression and allow for earlier therapeutic interventions. Keeping in mind that IgAN remains a highly heterogenous disease and that many questions still need to be answered, merging data from retrospective, risk factor studies and prospective treatment studies suggests that a disease-stratifying algorithm would be appropriate for disease management. Although the Oxford Classification of IgAN has been shown to provide earlier risk prediction and indirect evidence of IS responsiveness, it awaits validation in a prospective setting. With this available evidence, we propose an updated treatment algorithm and tried to delineate potential research directions. Ultimately, future challenges will be to identify noninvasive biomarkers to monitor disease activity and develop targeted, less toxic immunotherapies.

## Figures and Tables

**Figure 1 jcm-08-01584-f001:**
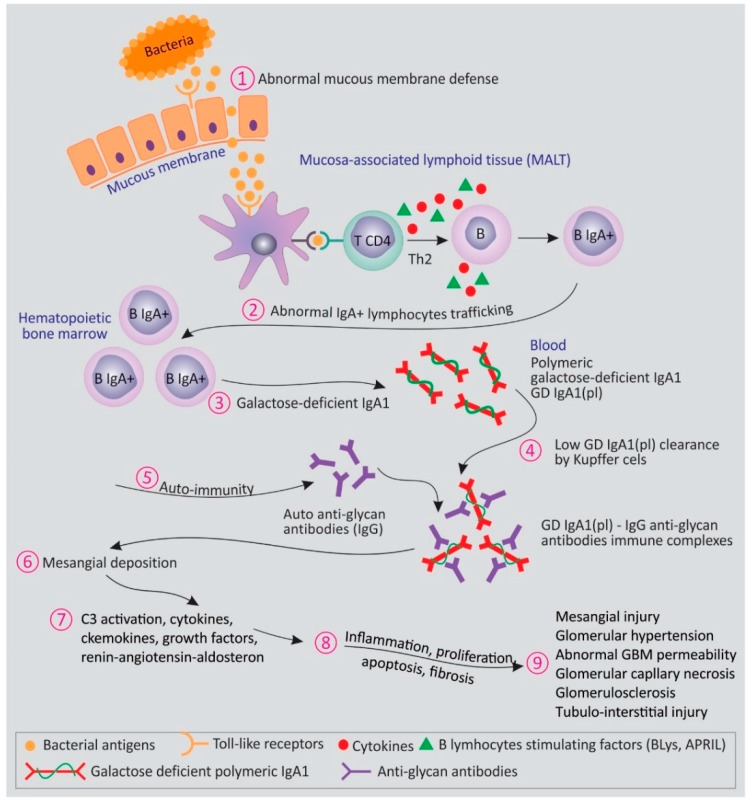
Pathogenesis of immunoglobulin A nephropathy (IgAN): the multihit hypothesis. The earliest event in IgAN pathogenesis is an increased production of circulating galactose-deficient (Gd)-IgA1, which is determined by several factors (genetic predisposition, abnormal mucosal immunity, mistrafficking of IgA1+ plasmablasts, decreased clearance of IgA1; Steps 1–4). This elicits an autoimmune response culminating with the production of antiglycan antibodies and the formation of immune complexes (Step 5). Subsequently, mesangial deposition and the activation of resident glomerular cells and of complement cascade will determine glomerular and tubulointerstitial injury (Steps 6–9).

**Figure 2 jcm-08-01584-f002:**
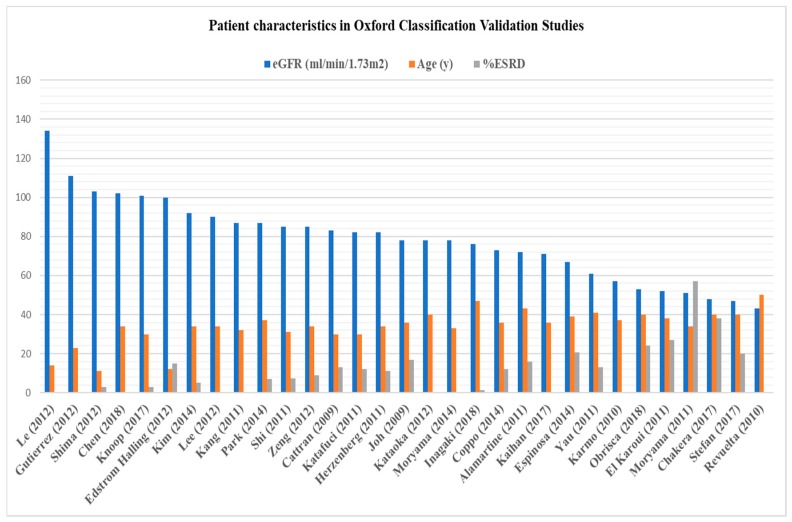
Baseline patient characteristics (estimated glomerular filtration rate (eGFR), age, 24-h proteinuria) in Oxford validation studies [7,42,43,44,45,46,47,48,49,50,51,52,53,54,55,56,57,58,59,60,61,62,63,64,65,66,67,68,69,70,71,72].

**Figure 3 jcm-08-01584-f003:**
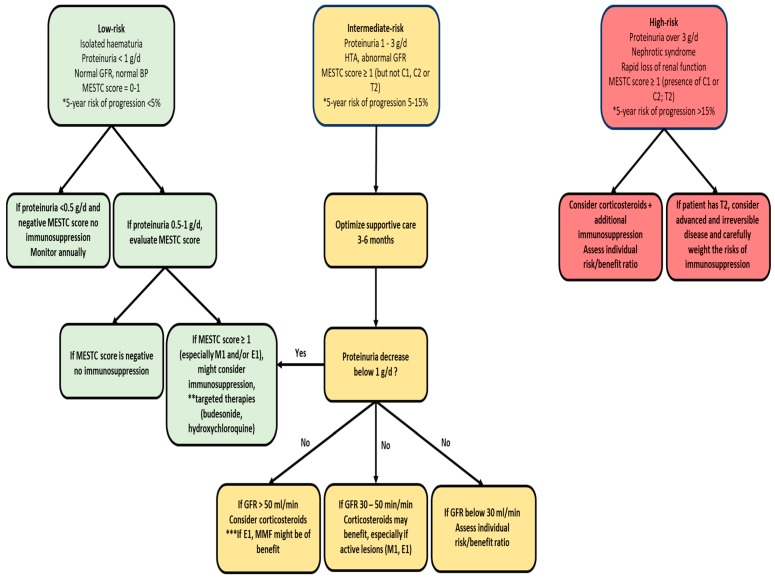
Proposed treatment algorithm for IgA nephropathy (adapted according to References [13,119,120]). * Evaluate the 5-year risk of progression (50% decline in estimated GFR or ESRD) according to the new international risk prediction tool [93]. ** Consider targeted therapies (budesonide, hydroxychloroquine) [105,121,122]. *** E1 lesion might be responsive to MMF, in addition to corticosteroids [77]. Abbreviations: GFR, glomerular filtration rate; BP, blood pressure; HTA, arterial hypertension; MMF, mycophenolate mofetil; ESRD, end-stage renal disease.

**Table 1 jcm-08-01584-t001:** Prospective, randomized, therapeutic trials on IgA nephropathy.

**A**						
**Trial**	**ACE inhibitors**		**Corticosteroids**			
	**Praga (2003) [103]**	**Coppo (2007) [104]**	**Shoji (2000) [96]**	**Katafuchi (2003) [97]**	**Hogg (2006) [99]**	**Koike (2008) [98]**
**Number of pts.**	44	66	21	90	96	48
**Serum creatinine (mg/dL)**	0.95 ± 0.2	-	0.74 ± 0.22	0.91 ± 0.22	-	1.04 ± 0.31
**eGFR (mL/min/1.73m^2^)**	100 ± 23	112 ± 21	106 ± 30	90 ± 26	114 ± 43	-
**Proteinuria (g/day)**	1.7–2	1.7 ± 0.7	0.75 ± 0.31	1.63 ± 1.53	1.4–2.2	0.93 ± 0.63
**RAAS blockade (%)**	52%	48%	0%	2%	52%	23%
**Treatment**	Enalapril vs. placebo	Benazepril vs. placebo	Prednisone vs. dipyridamole	Low-dose prednisone vs. dipyridamole	Prednisone vs. omega 3 fatty acids vs. placebo	Low-dose prednisone vs. dipyridamole
**Progression**						
**Definition**	50% increase in SCr	30% decrease in ClCr	-	ESRD	60% decrease in eGFR	-
**Proportion**	13% vs. 57%	3.1% vs. 14.7%	-	6.9% vs. 6.3%	6% vs. 25% vs. 13%	-
**∆GFR/year**	-	-	-	-	-	-
**Remission**						
**Definition**	-	Proteinuria below 0.5 g/d	Percentage decrease of proteinuria	Changes in urinary protein excretion from baseline	-	Changes in proteinuria from baseline
**Proportion**	-	40.6% vs. 8.8%	41% vs. 0%	−0.84 vs. 0.26	-	More decrease in steroid group
**Follow-up (months.)**	76	38	13	65	24	24
**Conclusion**	Positive	Positive	Positive	Negative	Negative	Positive
**B**						
**Trial**	**Corticosteroids**				
	**Pozzi (1999) [16,95]**		**Manno (2009) [15]**	**Lv (2009) [14]**	**Lv (2017) [19] TESTING Trial**	**Fellstrom (2017) [105] NEFIGAN Trial**
**Number of pts.**	86		97	63	262	149
**Serum creatinine (mg/dL)**	0.9–1.09		1.07 ± 0.3	1.1 ± 0.3	1.55 ± 0.6	-
**eGFR (mL/min/1.73m^2^)**	87–93		99 ± 27	101	59 ± 25	78 ± 25
**Proteinuria (g/day)**	1.8–2		1.5–1.7	2–2.5	2.4	1.2
**RAAS blockade (%)**	54%		100%	100%	100%	100%
**Treatment**	Corticosteroids vs. supportive treatment		Prednisone + ramipril vs. ramipril	Prednisone + cilazapril vs. cilazapril	Methylprednisolone vs. placebo	Budesonide 16 mg vs. budesonide 8 mg vs. placebo
**Progression**						
**Definition**	Doubling of SCr		Doubling of SCr or ESRD	50% increase in SCr	40% decrease in eGFR/ESRD/death	Percentage change from baseline of eGFR (9 months)
**Proportion**	2.3% vs. 30.2%		4.2% vs. 26.5%	3% vs. 24.1%	5.9% vs. 15.9%	0.6% vs. −0.9% vs. −9.8%
**∆GFR/year**	-		−0.56 vs. −6.17	-	−1.79 vs. −6.95	-
**Remission**						
**Definition**	Proteinuria below 0.5 g/d		-	-	Complete/partial remission at 12 months	Changes in urinary protein excretion from baseline (12 months)
**Proportion**	26% vs. 5% (after 1 year)		-	-	52.2% vs. 13.6%	−32% vs. −22% vs. 0.5%
**Follow-up (months)**	84		96	27	25	12
**Conclusion**	Positive		Positive	Positive	Possible renal benefit, excess infectious SAE	Positive
**C**						
**Trial**	**Mycophenolate mofetil**				
	**Maes (2004) [106]**		**Frisch (2005) [107]**	**Tang (2005) [108,109]**	**Hogg (2015) [110]**	**Hou (2017) [78]**
**Number of pts.**	34		32	40	52	176
**Serum Creatinine (mg/dL)**	1.42 ± 0.09		2.4 ± 0.96	1.59 ± 0.2	-	0.93
**eGFR (mL/min/1.73m^2^)**	71 ± 6 (inulin clearance)		39 ± 24	51 ± 4	100 ± 42	92
**Proteinuria (g/day)**	1.6		2.7	1.8	1.48	2.42
**RAAS blockade (%)**	100%		100%	100%	100%	24%
**Treatment**	MMF vs. placebo		MMF vs. placebo	MMF vs. placebo	MMF vs. placebo	MMF + low dose prednisone vs. full-dose prednisone
**Progression**						
**Definition**	25% decrease in inulin clearance		50% increase in SCr/ESRD	Doubling SCr/ESRD	-	ESRD
**Proportion**	33% vs. 15.3%		29% vs. 13%	15% vs. 50%	-	0% vs. 2.2%
**∆GFR/year**	−4.3 vs. −0.66		-	−1.12 vs. −3.81	−7 vs. 2.8 (at 6 months)	-
**Remission**						
**Definition**	-		Partial remission (50% reduction of proteinuria)	Partial remission (50% reduction of proteinuria)	Complete remission	Complete remission (undetectable proteinuria and stable renal function)
**Proportion**	-		17% vs. 13%	80% vs. 30%	None	48% vs. 53%
**Follow-up (months)**	36		24	72	24	12
**Conclusion**	Negative		Negative	Positive	Negative	Positive
**D**						
**Trial**	**Calcineurin inhibitors**				
	**Lai (1987) [111]**		**Kim (2013) [112]**		**Liu (2014) [113]**	**Xu (2014) [114]**
**Number of pts.**	19		40		48	96
**Serum creatinine (mg/dL)**	1.32		1.02 ± 0.28		1.01 ± 0.27	0.99 ± 0.23
**eGFR (mL/min/1.73m^2^)**	72 (creatinine clearance)		82 ± 22		80 ± 20	76 ± 24
**Proteinuria (g/day)**	3.35		1.3		2.88	2.04
**RAAS blockade (%)**	0%		50%		100%	100%
**Treatment**	Cyclosporine A vs. placebo		Tacrolimus vs. placebo		Cyclosporin A + medium-dose prednisone vs. full-dose prednisone	Cyclosporin A + medium-dose prednisone vs. full-dose prednisone
**Progression**						
**Definition**	-		-		25% decrease of eGFR	-
**Proportion**	-		-		9% vs. 0%	-
**∆GFR/year**	-		-		-	-
**Remission**						
**Definition**	50% reduction of proteinuria		Percentage decrease of proteinuria		Complete remission	Complete remission
**Proportion**	77% vs. 0%		52% vs. 17%		50% vs. 45.8%	52% vs. 21%
**Follow-up (months)**	7		4		36	12
**Conclusion**	Negative (renal function deterioration in CSA group)		Positive		Positive	Positive
**E**						
**Trial**	**Azathioprine**		**Rituximab**		**Cyclophosphamide/Azathioprine**	
	**Pozzi (2010) [115]**		**Lafayette (2017) [116]**		**Ballardie and Roberts (2001) [117]**	**Rauen (2015) [18] STOP trial**
**Number of pts.**	207		34		38	337 (Run-in phase) 162 (Trial phase)
**Serum creatinine (mg/dL)**	1.2 (1–1.6)		1.4 (0.8–2.4)		-	1.5 ± 0.6
**eGFR (mL/min/1.73m^2^)**	66 (48–87)		49 (30–122)		-	61 ± 27
**Proteinuria (g/day)**	2 (1.5–3)		2.1 (0.6–5.3)		4.25	2.2 ± 1.8
**RAAS blockade (%)**	90%		100%		-	100%
**Treatment**	Corticosteroids vs. corticosteroids + azathioprine		Rituximab vs. placebo		Corticosteroid + cyclophosphamide followed by azathioprine vs. placebo	Pozzi/Ballardie regimen vs. placebo
**Progression**						
**Definition**	50% increase in SCr		25% decrease of eGFR		5-year renal survival	eGFR decrease ≥ 15 mL/min/1.73m^2^
**Proportion**	11.3% vs. 12.9%		1/17 vs. 0/17		72% vs. 5%	26% vs. 28%
**∆GFR/year**	-		-		−1.07 vs. −5.12	−1.4 vs. −1.6
**Remission**						
**Definition**	Percentage decrease of proteinuria		Partial remission		-	Full clinical remission
**Proportion**	49.9% vs. 44.8%		3/16 vs. 3/15		-	17% vs. 5%
**Follow-up (months.)**	59		12		Up to 72	36
**Conclusion**	Negative		Negative		Positive	More complete remissions in steroid groups, no difference in renal function decline

Abbreviations: ACE inhibitors, angiotensin-converting enzyme inhibitor; pts, patients; eGFR, estimated glomerular filtration rate; RAAS, renin-angiotensin-aldosterone system; ESRD, end-stage renal disease; MMF, mycophenolate mofetil; SAE, serious side effects; CSA, cyclosporine A.

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
