# Peer review of "Has The Time Arrived to Refine The Indications of Immunosuppressive Therapy and Prognosis in IgA Nephropathy?"

_jcm, 2019, doi:10.3390/jcm8101584_

Round 1

Reviewer 1 Report

Authors discussed issues concerning the immunosuppressive treatment of IgAN. This review is concise and well written. There are only a few comments.

Do authors have a direct answer to your own article topic? If so, please do address it clearly in your conclusion, which can be expanded. This article chiefly focused on immunosuppressive therapy. Authors can consider to mention other treatment strategies, for example, the role of tonsillectomy to remove the source of nasopharyngeal activation of mucosal immunity. Though not widely recognized by nephrologists worldwide, tonsillectomy may have its place in the treatment of IgAN. For non-invasive IgAN marker,Gd-IgA1 is very promising, as mentioned in your text. I suggest that authors might consider to include the Kidney International paper (2018; 93(3):700-705) and describe the finding in your text.

Author Response

Authors discussed issues concerning the immunosuppressive treatment of IgAN. This review is concise and well written. There are only a few comments.

Thank for your comments and suggestions. We have revised the manuscript based on your recommendations. Together with revised manuscript here is our response.

Point 1. Do authors have a direct answer to your own article topic? If so, please do address it clearly in your conclusion, which can be expanded.

Response 1. We have updated the conclusion section according to your recommendations

Point 2. This article chiefly focused on immunosuppressive therapy. Authors can consider to mention other treatment strategies, for example, the role of tonsillectomy to remove the source of nasopharyngeal activation of mucosal immunity. Though not widely recognized by nephrologists worldwide, tonsillectomy may have its place in the treatment of IgAN.

Response 2. We have introduced a discussion on the role of tonsillectomy in the management of IgA Nephropathy.

Point 3. For non-invasive IgAN marker,Gd-IgA1 is very promising, as mentioned in your text. I suggest that authors might consider to include the Kidney International paper (2018; 93(3):700-705) and describe the finding in your text.

Response 3. We have updated the discussion about Gd-IgA1 according to your suggestion and we included the recommended paper and additional references to better underlie the possible utility of this marker for future disease management.

Reviewer 2 Report

The authors have compiled an interesting review on the indications of immunosuppressive therapy and prognosis of IgA nephrotherapy. There are certain sections those needs to be improved/ modified:

Include a scope section that should discuss the need of compiling this review article, aspects covered, literature search that includes search engines and keywords used. Include a section on future direction. Add a figure on the line of suggested treatment for IgA nephrotherapy. The authors can take example from the book Davidson's Principles and Practice of Medicine.

Author Response

The authors have compiled an interesting review on the indications of immunosuppressive therapy and prognosis of IgA nephropathy. There are certain sections those needs to be improved/ modified:

Thank for your comments and suggestions. We have revised the manuscript based on your recommendations. Together with revised manuscript here is our response.

Point 1. Include a scope section that should discuss the need of compiling this review article, aspects covered, literature search that includes search engines and keywords used.

Response 1. We included a paragraph mentioning the scope of this review and aspects covered. However, given the fact that this review is not intended to be a systematic review/metanalysis and more of a general review article, we only selected references and studies that we considered relevant for our view of IgA nephropathy pathogenesis/prognosis/treatment. As such, we do not mention the search engine/keywords used because this approach is more accurate for a systematic review/metaanalysis, while we used the typical references mentioned in this type of review.

Point 2. Include a section on future direction.

Response 2. We have added a future direction section.

Point 3. Add a figure on the line of suggested treatment for IgA nephropathy. The authors can take example from the book Davidson's Principles and Practice of Medicine.

Response 3. We have added a figure regarding the suggested treatment of IgA nephropathy. Additionally, we have added a couple of references to support our treatment algorithm and the most recent studies regarding hydroxychloroquine and budesonide. Also, we updated the risk evaluation of IgA Nephropathy with the new international risk prediction tool (Barbour S et al. Evaluating a New International Risk-Prediction Tool in IgA Nephropathy. JAMA 2019)

Round 2

Reviewer 2 Report

The authors have answered the comments satisfactorily.